# ADAPTIVELY LABELING VISION DATASETS
# VIA INSTANCE-LEVEL RETRIEVAL

## ABSTRACT

Human annotations are the backbone of modern computer vision, but they are increasingly recognized as an inefficient resource. They typically capture only a single, fixed view of the rich visual information present in images. How can we move toward datasets that are labeled adaptively, rather than exhaustively by hand? We propose Instance-Level Retrieval, a method for adaptively building object detection datasets from large collections of unlabeled images. Given just a handful of seed examples, our method automatically finds and labels relevant training data by comparing self-supervised object representations. Starting from a small, subset of Pascal VOC, we demonstrate that it is possible to retrieve a high quality set of images. In experiments that control data scale, models trained on our adaptively-labeled data exceed the performance of training on the original Pascal VOC human annotations with a $0.08$ mAP improvement. We use our retrieval method on out of distribution unlabeled images derived from ImageNet-1K, showing that our method can successfully find high quality exemplars for fixed image classes matching the Pascal VOC training set. Training on this expanded training set leads to an additional $0.105$ mAP improvement over baseline. Finally, we show that our methodology is also useful for filtering and selecting high-quality subsets of human-annotated data, yielding a $0.037$ mAP gain compared to uniformly sampled subsets.

## 1 INTRODUCTION

The current paradigm in modern computer vision is to curate a fixed set of human annotations tailored to a particular downstream task such as object detection. Annotators are typically provided with labeling instructions—often restricted to a predetermined set of visual categories—and asked to localize objects of interest within a small, fixed pool of images (Deng et al., 2009; Gupta et al., 2019; Lin et al., 2014; Everingham et al., 2010). While effective, this paradigm does not scale well to the vast amounts of image data available on the internet, where the key challenge lies in retrieving high-quality and task-relevant exemplar images. Moreover, these annotations generally capture only a single, fixed view of the rich visual information inherent in images.

In this work, we ask: how can we construct computer vision datasets that are adaptively labeled to meet evolving application demands? To address this question, we introduce Instance-Level Retrieval, a method inspired by Retrieval-Augmented Generation (Lewis et al., 2020) in language models. Given only a handful of seed examples, our approach adaptively builds a training set by retrieving the most relevant instances from large collections of unlabeled images, guided by self-supervised object representations.

Our method, Instance-Level Retrieval, operates in three stages. First, we use a pretrained region proposal network to extract candidate object instances from a large pool of unlabeled images. Second, we represent each candidate with features from a state-of-the-art self-supervised encoder. Finally, given a small set of seed images with bounding box annotations, we retrieve the top-$k$ most relevant candidates by nearest-neighbor search in the representation space. These candidates are then assigned labels based on their similarity to the seeds, yielding an adaptively labeled dataset without requiring additional manual annotation..

We evaluate this approach by extracting training samples for Pascal VOC (Visual Object Classes) (Everingham et al., 2010) from just a handful of key examples. In experiments that control for data

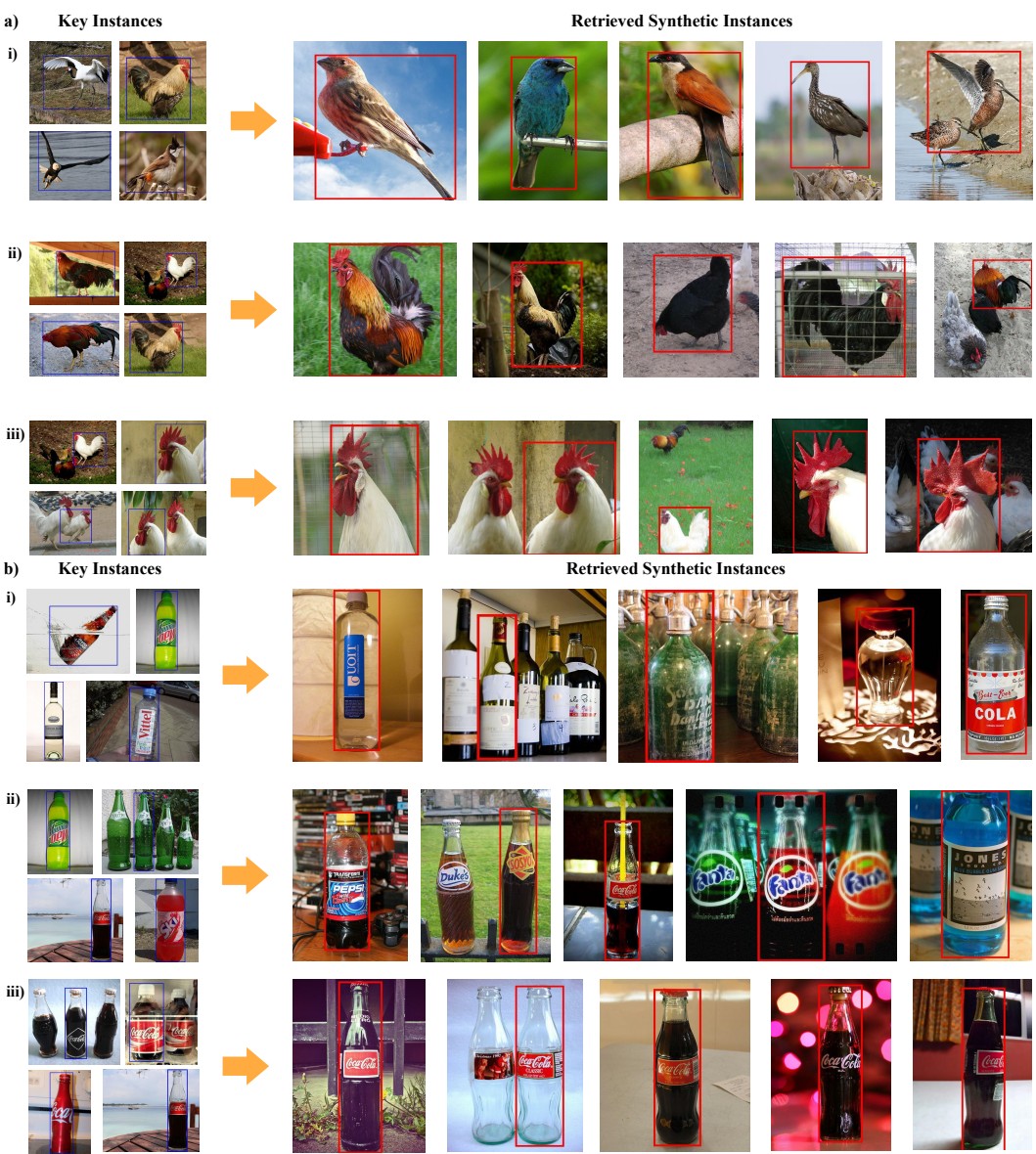

Figure 1: **Capabilities of the adaptive labeling algorithm**: We can choose retrieval keys and set thresholds within the retrieval algorithm to obtain different levels of granularity in the obtained adaptively labeled instances. **(i)** We start with keys belonging to different classes in the ImageNet-1K dataset, modifying the algorithm to also factor in variability, the outputs represent a generalization of the selected anchors.We see different kinds of birds in **(a)** while **(b)** shows varying bottles. **(ii)** We now choose keys belonging to the same class in the ImageNet-1K dataset. Using the default algorithm as described in Section 3.4 the outputs capture other instances of the same class. We capture cockerels in **(a)** while **(b)** shows soda bottles. **(iii)** We proceed to visually similar keys with more granularity than the ImageNet-1K classes. By increasing the *Semantic IoU* sensitivity in the default algorithm we are able to get more fine-grained results than the available human-annotated classes. We get a sub-class of white cockerels in **(a)** while **(b)** shows specifically Coca-Cola bottles.

scale, models trained on our adaptively labeled data outperform those trained on the original human annotations, achieving a +0.08 mAP improvement. Using adaptive labeling to transfer labels from PASCAL VOC to unlabeled ImageNet-1K(Deng et al., 2009) images, we are able to extend the PASCAL VOC training set and achieve an average 0.105 mAP improvement over models trained only using the base training set. Finally, we demonstrate that our methodology can also be used for dataset distillation: filtering human-annotated data into smaller, higher-quality subsets yields a 0.037 mAP improvement compared to uniformly sampled subsets.

## 2 RELATED WORK

**Adaptive Computer Vision.** Computer vision researchers have worked on adaptivity in prior works, focusing primarily on building adaptive models, rather than adaptive datasets. For example, Large Vision Models (Bai et al., 2023) are a class of autoregressive model that can perform new tasks via few-shot in-context learning. Similarly, detection models like Owlv2 (Minderer et al., 2022; 2024) can have vision encoders attached, which allow for detection of novel objects from a single example image. Perhaps the most relevant works to consider adaptive datasets are those in domain adaptation (Csurka, 2017), which consider target domain with a potentially significant domain shift from a static source domain. The main difference between this work and domain adaptation works lies in the formulation of the dataset. Our retrieval step dynamically rebuilds the data-distribution in its entirely based on images from the target domain, whereas previous works in domain adaptation often focus on loss function and model augmentations to mitigate domain shift (Csurka, 2017).

**Retrieval Augmented Generation.** Retrieval-Augmented Generation has recently emerged for language models as a way to mitigate hallucinations (Lewis et al., 2020). By allowing the model to augment its context with a relevant document retrieved from a larger source of relevant knowledge, language models can be extended to knowledge from beyond their initial training data. RAG has recently been extended to Diffusion models (Luo et al., 2024) to extend their capabilities to novel artistic concepts and styles created by users that such models were not initially trained to generate. Recent work has begun shifting adaptivity efforts from models to the datasets themselves through automatic and minimally supervised curation pipelines. For example, Automatic Data Curation (Vo et al., 2024) uses repeated hierarchical k-means clustering over large unlabeled pools to automatically build balanced, diverse, "ImageNet-quality" pretraining sets, outperforming raw uncurated data and rivaling expert-curated collections with no human labels. Retrieval is especially promising in data-centric adaptivity.

**Synthetic Data.** Synthetic data in computer vision has recently emerged as an effective strategy for training data-efficient models with limited real data (Trabucco et al., 2023; He et al., 2023; Wu et al., 2023; Azizi et al., 2023). For object detection, synthetic data methods have explored generating synthetic images alongside their labels (Wu et al., 2023), and using pretrained object detectors to weakly annotate unlabeled images for training larger models (Minderer et al., 2024). Precision at Scale (de Vera et al., 2024) leverages LLMs, generative models, and CLIP filtering to synthesize domain-specific datasets that can even surpass ImageNet-pretrained baselines. Together, these works highlight adaptive curation—retrieved, clustered, or generated on demand—as a new parallel to adaptive model design. However, these approaches ultimately do not solve the adaptivity problem. When the required visual task changes—often the case in real-world settings—synthetic data must often be re-generated.

## 3 METHODOLOGY

Our setup is as follows: We consider a small set of hand labeled anchor instances made up of images and bounding box annotations and a much larger set of unlabeled images. First, we find candidate instances in all of the unlabeled images using standard object detection models (Section 3.1). Next, we measure the similarity between our anchor instances and unlabeled candidate instances with our *Semantic Intersection over Union (Semantic IoU)* (Sections 3.2, 3.3). Finally, we filter for high-quality unlabeled instances and return the newly labeled instances for downstream training (Section 3.4).

### 3.1 BOUNDING BOX PROPOSAL

For a given unlabeled image, the initial step involves generating a set of bounding box proposals. We achieve this using the Owlv2ForObjectDetection model (Minderer et al., 2024), which produces candidate bounding boxes that are subsequently filtered based on their objectness score. Each resulting bounding box, along with its corresponding image, is referred to as an instance.

To refine the set of proposed instances, we apply Non-Maximum Suppression (NMS) to eliminate bounding boxes with excessive overlap. We compute the pairwise Intersection over Union (IoU) for all bounding boxes within a given image. If the IoU between any two boxes exceeds a predefined threshold, we discard the instance with the lower objectness score. We use a relatively high threshold of 0.8 to ensure we only remove duplicate detections of the same object, without eliminating distinct objects that may overlap. A stricter threshold is then applied during class assignment to enforce unique instances.

### 3.2 BAG-OF-FEATURES ENCODING

Once the set of instances is obtained, we process each instance using the Segment Anything Model 2 (SAM2) (Kirillov et al., 2023) to generate a segmentation mask over the principal object. Simultaneously, we pass the image through the DINOv2 model (Oquab et al., 2024) to extract its patchwise feature map. We then retrieve the patchwise features from this feature map that correspond to the computed mask for each instance, ensuring that only the relevant regions are utilized for further processing. For each candidate instance, this process results in a set of patch features that were contained within the bounds of the segmentation mask. This set is defined below.

$$X = \{\vec{\mathbf{x}}_1, \vec{\mathbf{x}}_2, \cdots, \vec{\mathbf{x}}_N\} \ \text{ st } \ \vec{\mathbf{x}}_i \in \mathcal{R}^D \tag{1}$$

Here, each $\vec{\mathbf{x}}_i$ corresponds to a single patch feature selected from the final layer predictions of Dinov2 (Oquab et al., 2024). Although $X$ is conceptually a set since patch features are unordered, we treat it as a matrix with $N$ rows and $D$ columns with $N$ rows and $D$ columns, where $N$ is the number of patch locations within the segmentation mask for the object, and $D$ is the dimensionality of the Dinov2 feature space.

### 3.3 SEMANTIC IoU CALCULATION

Equipped with a self-supervised representation for objects based on a set of patch-level features, we develop a similarity metric to compare two bag-of-features representations. Given two such representations $X$ and $Y$ defined in the previous Section 3.2, we compute their *Semantic Intersection Over Union* (*Semantic IoU*) as follows. We first normalize all vectors within each set to unit length. Then, we apply the hungarian matching algorithm (Kuhn, 1955) to pair vectors from the set $X$ to the set $Y$ in order to maximize cosine similarity of paired vectors.

$$\hat{P} = \arg \max_{P \in S_{N,M}} \text{Tr}\left(X(PY)^T\right) \tag{2}$$

The matrix $P^*$ found by the hungarian matching algorithm is a non-square permutation-like matrix from the set of non-square permutation-like matrices from $N$ elements to $M$ elements, noted $S_{N,M}$ in Equation 2. The matrix is optimized to maximize the sum of cosine similarities of paired vectors, induced by $P^*$, accomplished above using the matrix interpretations for $X$ and $Y$. Here we assume that $X$ has $N$ elements as defined previously, and $Y$ is defined similarly to $X$, but with $M$ elements. With the optimal match between the sets, we compute *Semantic IoU* with the following.

$$\text{Semantic IoU} = \underbrace{\text{Tr}\left(X(\hat{P}Y)^T\right)}_{\textbf{intersection}} \Big/ \underbrace{\left(N + M - \text{Tr}\left(X(\hat{P}Y)^T\right)\right)}_{\textbf{union}} \tag{3}$$

Intuitively, this metric favors objects that are visually similar (have many patch features with high cosine similarity), and are of comparable size (the number of patches).

### 3.4 INSTANCE-LEVEL RETRIEVAL

We generate adaptively labeled data using a matching algorithm inspired by Retrieval-Augmented Generation (RAG) (Lewis et al., 2020). We begin by selecting a retrieval key, which is a human-

labeled instance—referred to as the anchor instance—for which we aim to generate similar instances from unlabeled data.

After generating bounding box proposals for the unlabeled data, we extract bag-of-features encodings for both the generated instances and the anchor instance. We then compute the *Semantic IoU* between the anchor instance and all candidate instances.

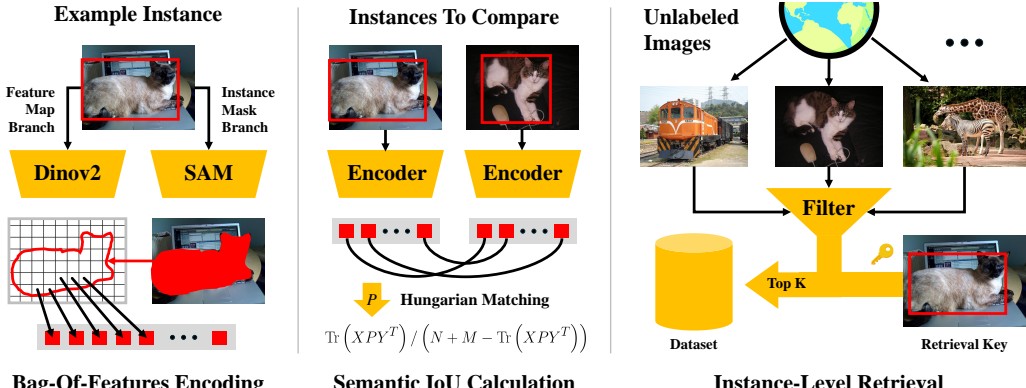

Figure 2: **Overview of instance-level retrieval:** Starting from a large unlabeled set of images, we employ pretrained vision models to discover objects for training based on a handful of examples. Our method finds the most suitable instances by comparing self-supervised representations for objects using a proposed *Semantic IoU* metric sensitive to visual appearance and structural similarity.

Instead of directly selecting the top $K$ instances most similar to a given anchor instance, we initially retain the top $10 * K$ instances based on their similarity scores. We filter this superset by applying another, more stringent round of NMS as described in Section. 3.1.

For each of the filtered candidate instances, we maintain a comprehensive assignment that documents the corresponding anchor instance(s) that retrieved it, the associated IoU value, and the label of the anchor(s). Once we have iterated over all anchor instances, this record serves as a mapping from the extracted instances to all the anchor instances for which it was deemed among the $K$ nearest neighbors. We use this mapping to perform targeted filtering to obtain a final, high quality dataset.

We consider two filtering methods designed to filter out low quality and confusing candidate instances. Firstly, we filter out candidate instances below a minimum number of assigned anchor instances. This removes instances that are only match to a single or few anchor images in a given class. Next, we filter out candidate instances for which the assigned anchor instances have conflicting labels. Specifically, we drop candidate instances where no single anchor label constitutes a majority of matched anchor instances. Extracted instances exceeding both thresholds are assigned the majority label, ensuring high quality and classification robustness in ambiguous cases.

## 4 RESULTS

We evaluate our approach through carefully controlled experiments that highlight its ability to retrieve and label high-quality class examples from a small set of anchor instances. We test our retrieval method on unlabeled data with the same distribution as the anchor labels, as well unlabeled data with a different distribution from our anchor instances. Our results show that the method can retrieve and label high-quality examples in both cases, increasing performance when compared to baselines.

**Dataset details:** We use PASCAL VOC 2012 (Everingham et al., 2010) and ImageNet-1K (Deng et al., 2009) for our experiments, using the latter only as a source for unlabeled images. The PASCAL VOC 2012 contains 5717 training images and 5823 validation images, with 20 categories. We use these 20 object classes as target labels for all of our experiments. All results are reported on the PASCAL VOC 2012 validation set.

We use a sampled subset of ImageNet-1K; 250 images from each of the 1000 classes, to act as the set of unlabeled image where we extract our instance proposals from for the transfer labeling experiment in Section. 4.2.

**Training details:** We use $K = 10$ for the instance-level retrieval step for the in-domain labeling and $K = 5$ for transfer labeling. Since the former works within the PASCAL VOC dataset, we have less data to work with and require more assignments, in the latter since our candidates are from the ImageNet-1K we can afford having stricter assignment policies with more available images and thus instances. Our selection choices are explained in Section. 5.2.

For all object detection models used for evaluation in Section. 4, we train a YOLO11m model from scratch for 300 epochs with batch size 16 with an image size set to 640 pixels. We use SGD as our optimizer with a learning rate of 0.01. Our chosen performance metric is the highest Mean Average Precision (mAP) achieved by the model on the PASCAL VOC validation set. It is computed as the mean of the Average Precision (AP) across all object categories. The reported mAP@50 and mAP@50-95 is averaged over 4 seeds and correspond to the AP averaged over IoU thresholds of 0.50 and the range from 0.50 to 0.95, respectively.

## 4.1 IN-DOMAIN DATA

**Experimental setup:** To demonstrate the training efficiency gained through the proposed adaptive labeling algorithm, we compare the performance of an object detection model trained on the original PASCAL VOC human annotations with a model trained on an adaptively labeled dataset derived from the same source. To construct the datasets, we follow the standard pipeline. Using unlabeled versions of the images in the training set as the source and a tiny set of ground-truth annotations as retrieval keys (10 and 40 anchors for 1000 and 4000 instances respectively) , we carry out the algorithm described in Section. 3.4. Once we have obtained label assignments for the adaptively extracted instances, we select the top $N$ instances for each class with highest assigned *Semantic IoU*. To ensure a fair comparison, we subsample $N$ instances per class from the original training set, matching the dataset size of the adaptively labeled version.

Table 1: In-domain adaptive labeling

| Instances | Labeling | mAP@50 | mAP@50-95 |
|---|---|---|---|
| 1000 | Adaptive | $0.250 \pm 0.004$ | $0.154 \pm 0.003$ |
| | Human | $0.166 \pm 0.008$ | $0.091 \pm 0.004$ |
| 4000 | Adaptive | $0.452 \pm 0.007$ | $0.311 \pm 0.007$ |
| | Human | $0.361 \pm 0.010$ | $0.229 \pm 0.009$ |

**Understanding the results:** Table. 1 presents a comparison between models trained on adaptively labeled data and those trained on original human-annotated data, on two dataset scales; 1000 and 4000 total instances across all classes. In this setup, all classes have a uniform number of training samples to ensure a fair evaluation. Our method outperforms the baseline of training on a random subset of human labeled datapoints from the Pascal VOC dataset, showing that our method successfully retrieves and labels high quality training points from the overall training set.

For the smaller subset of 1000 instances, the model trained on adaptively labeled data achieves an average 0.0735 improvement for mAP@50 and mAP@50-95 over human labeled data. At the larger scale of 4000 instances, both labeling strategies benefit from increased data, but the adaptive approach continues to outperform human annotations by an average 0.0865 mAP. Consistent improvements in different metrics and dataset sizes, averaging 0.08 mAP, show that the adaptive labeling algorithm is effective in creating high-quality training data that increases accuracy on our held out validation set.

## 4.2 TRANSFER LABELING

**Experimental setup:** Next, we showcase our algorithms ability to retrieve and label high quality class instances from a larger unlabeled dataset with a different distribution than our train set. In this

experiment, we use our method to retrieve ImageNet-1K (Deng et al., 2009) instances using class labels and anchors from the PASCAL VOC dataset. For transfer labeling, we first extract candidate instances from ImageNet-1K images. The human-annotated PASCAL VOC training instances serve as retrieval keys, against which we use the same algorithm to obtain adaptively labels for matched ImageNet candidates.

We implement two distinct experimental configurations; a purely adaptive setup and a mixed setup. In the purely adaptive configuration, we sample $x$ times the number of human-labeled instances in the PASCAL VOC training set for each class from the adaptively labeled instances, and only use the retrieved instances for training. The mixed configuration combines adaptively labeled instances at scale factor $x$ with human-labeled instances at scale factor $y$.

The choice of choosing a scale factor is important since PASCAL VOC is a highly unbalanced dataset. By choosing a scale with respect to class we ensure that the retrieved data imbalance matches the original PASCAL VOC dataset for a fair comparison.

Table 2: Transfer adaptive labeling

| Adaptive scale | Human scale | mAP@50 | mAP@50-95 |
|---|---|---|---|
| 1.0× | - | $0.548 \pm 0.011$ | $0.427 \pm 0.014$ |
| 1.5× | - | $0.579 \pm 0.014$ | $0.446 \pm 0.018$ |
| 0.5× | 0.5× | $0.617 \pm 0.005$ | $0.455 \pm 0.004$ |
| 1.0× | 0.5× | $0.657 \pm 0.005$ | $0.498 \pm 0.006$ |
| 0.5× | 1.0× | $0.689 \pm 0.004$ | $0.520 \pm 0.006$ |
| 1.0× | 1.0× | $0.711 \pm 0.004$ | $0.550 \pm 0.005$ |
| - | (Baseline) 1.0× | $0.668 \pm 0.012$ | $0.488 \pm 0.014$ |

**Understanding the results:** The experimental results in Table. 2 showcase the effectiveness of transfer labeling, either by itself or appended to human annotations, and provide a baseline of training with the complete PASCAL VOC training set. Despite not using human annotations for the purely adaptive approach, it delivers competitive performance, showcasing the algorithm's robust capability to capture and transfer semantic information from anchors.

When examining the mixed configuration where both adaptive and human labels are used together, the data shows consistent performance improvements across different scale combinations. As shown in the data, when human labels (at 1.0× scale) are supplemented with adaptive labels (at 1.0× scale), the model achieves a mAP@50 of 0.711 and mAP@50-95 of 0.550, which provides an average 0.105 mAP improvement over baseline.

### 4.3 HUMAN DATA CURATION

**Experimental setup:** Our pipeline can also be manipulated to obtain a high-quality subset of instances from given human-annotated data. To do this, we skip extracting instances and instead use the human-annotated PASCAL VOC training set instances as both the retrieval keys and the candidate instances. We modify the algorithm from Section. 3.4 by restricting *Semantic IoU* calculation to candidates with the same ground-truth label. Furthermore, we skip over the top $K$ selection and the NMS step, since we have well-defined bounding boxes. We continue to maintain a record for this redefined assignment step. To obtain the dataset, for each class, we choose the top $N$ instances that have the highest average *Semantic IoU*.

**Understanding the results:** Table. 3, compares models trained on two scales of subsets of PASCAL VOC training data, where we select the subset by filtering using the proposed data curation algorithm or by random sampling. For the smaller subset of 1000 instances, the model trained on curated data achieves an average 0.039 improvement for mAP@50 and mAP@50-95 over human labeled data. At the larger scale of 4000 instances, there is a diminished effect of sampling, thus the extent of improvement drops slightly to an average of 0.034 mAP. With an overall average improvement of 0.0365 mAP, the proposed curation strategy is effective in distilling a high-quality subset from existing human-annotated data.

Table 3: Human data curation using *Semantic IoU*

| Instances | Selection | mAP@50 | mAP@50-95 |
|---|---|---|---|
| 1000 | Filtered | $0.210 \pm 0.004$ | $0.125 \pm 0.002$ |
| | Random | $0.166 \pm 0.008$ | $0.091 \pm 0.004$ |
| 4000 | Filtered | $0.399 \pm 0.002$ | $0.259 \pm 0.004$ |
| | Random | $0.361 \pm 0.010$ | $0.229 \pm 0.009$ |

## 5 ABLATIONS

### 5.1 LABEL ASSIGNMENT PERFORMANCE

**Experimental setup:** We examine the performance of the initial candidate to anchor assignment in the RAG-inspired instance-level retrieval algorithm in Section. 3.4. Instead of processing the entire dataset through the complete pipeline to generate bounding boxes, we utilize the pre-existing bounding boxes provided in the PASCAL VOC dataset. We iterate through the validation set, treating each instance as an anchor. For each anchor, we compute the *Semantic IoU* against all instances in the training set and identify the top $K$ most similar instances. The anchor is then assigned the label corresponding to the mode of these $K$ nearest instances. To assess the label assignment performance we define two metrics. *Accuracy* is defined as the proportion of validation instances assigned a label that matches their ground truth. *Consistency* measures how many of the $K$ selected instances share the same ground-truth label as the anchor instance.

Table 4: Accuracy and consistency for different values of $K$

| $K$ | Accuracy | Consistency |
|---|---|---|
| 1 | 0.941 | — |
| 5 | 0.951 | 0.927 |
| 10 | 0.953 | 0.915 |
| 15 | 0.953 | 0.906 |
| 20 | 0.951 | 0.898 |

**Understanding the results:** We present the accuracy and consistency metrics for different values of $K$ in the RAG-inspired instance-level retrieval algorithm in Table. 4. Increasing $K$ provides a slight improvement in accuracy that plateaus quickly. The consistency metric shows a clear pattern of diminishing returns. This suggests that as K increases, the algorithm begins to include less semantically similar instances, potentially introducing noise into the label assignment process. These findings indicate that K=5 or K=10 provides an optimal balance between precision and consistency for the RAG-inspired retrieval algorithm.

### 5.2 MAJORITY FRACTION SENSITIVITY

**Experimental setup:** We consider the transfer labeling experiment in Section. 4.2, and investigate the impact of the majority label fraction threshold for candidate instances on the performance of object detection models, as described at the end of Section. 3.4. For each dataset configuration, we evaluate a range of majority fraction thresholds from 0.50 to 0.90. The lower bound of the threshold is chosen to ensure a sufficient level of confidence in the correctness of the assigned majority label, particularly given the challenges posed by out-of-domain classes that may exhibit high visual similarity to the target categories.

**Understanding the results:** There seems to be no clear pattern between object detection performance and the majority label fraction threshold. The best performance for dataset configurations varies independently, however the difference in results for a given setup is marginal in the stipulated threshold range as compared to the differences in overall mAP@50 across data mixtures.

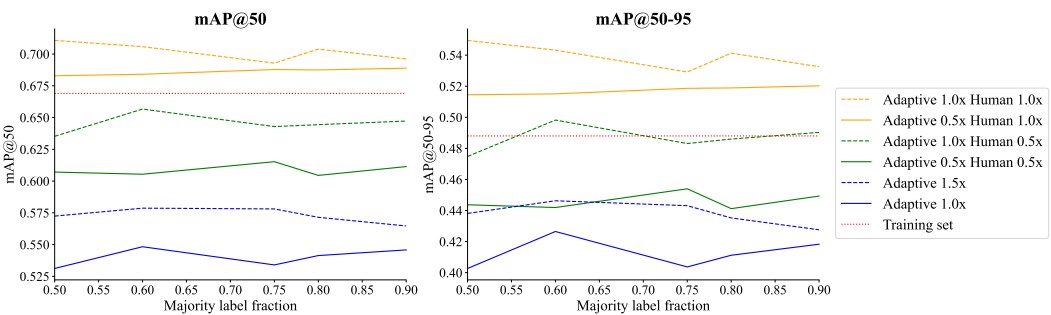

Figure 3: **mAP values at different majority label fraction thresholds for different setups** : We plot the performance of object-detection models in terms of mAP@50 and mAP@50-95 as we increase majority label fraction threshold, for different dataset configurations chosen in Section 4.2.

## 6 CONCLUSION

In this work, we introduced an adaptive labeling framework that retrieves semantically similar image regions from large unlabeled pools using a novel Semantic IoU similarity. Across PASCAL VOC and ImageNet experiments, our approach consistently yields higher mAP than static human annotations at comparable scales, supports cross-domain transfer, and enables more efficient human-in-the-loop curation by adaptively labeling unlabeled datapoints given a small set of anchor instances.

These results suggest that adaptive datasets constructed via retrieval may complement or even surpass traditional static benchmarks, opening new directions for data-centric machine learning for unseen or low-data classes. Moreover, the method holds promise for supporting resource-constrained training scenarios by selecting higher-quality subsets, thereby enabling effective model development at smaller computational scales.

While our method depends on the quality of anchor annotations and feature representations, future work could explore integrating stronger vision encoders, scaling retrieval to web-scale corpora, and extending the approach to domains such as video or 3D perception. We hope this work stimulates broader investigation into adaptive dataset design as a critical lever for advancing generalization in machine learning.

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

## A  APPENDIX

## B  LIMITATIONS & SAFEGUARDS

Automatic labeling of computer vision datasets has unique challenges and risks. One important limitation is that content filtering may be become reliable if a human annotator does not make the final decision to include a data point in a dataset. Our pipeline can allow for automatic annotation of freeform image datasets for object detection, and without a content filter, may allow for images with harmful content to enter training. Before our method can be used in the wild, the inclusion of a content filter will be important. Our main safeguard to prevent this issue in this paper is to restrict our focus to annotating existing datasets that have been deemed as safe by the research community.

Another limitation of this paper is the quadratic computational complexity to retrieve top instances from unlabeled images. As the size of an unlabeled dataset grows, the cost of retrieving data can grow beyond a reasonable budget, and may grow to the point where it is more expensive to apply our method that to employ human labels. To combat this, using our method in a product environment likely requires applying it to only a subset of unlabeled images, selected by a dataset partitioning strategy, such as by filtering images based on similarity to a caption according to CLIP.

## C  ETHICAL CONSIDERATIONS

All datasets used in this work were obtained via official releases from papers that took the necessary steps to make sure that images were obtained ethically. While the data is in this work is obtained ethically, our method allows users to annotate arbitrary images from the internet with minimal effort. It is up to the discretion of the final user to ensure that they obtain images via ethical means, as our method is purely a data annotation tool. In our code release, we will provide a safety filter to aid in the detection of harmful content in unlabeled images, and we recommend this filter be used

## D  BROADER IMPACTS

Our method reduces the annotation cost for object detection datasets, and allows for the creation of datasets for object detection from a handful of examples. This can displace human data annotators who work on platforms like mechanical turk once data from our pipeline matches their annotation quality, at a cheaper cost. In addition, due to the ease of dataset creation, object detectors can be created from previously hard-to-annotate sources, like video feeds, and streams, allowing detectors for specific people to be derived from these, which may harm user privacy if employed this way.

Another impact of moving towards adaptively labeled vision datasets is that the amount of tasks that can be derived from a single dataset increases. Previously unlabeled objects in datasets like ImageNet, and COCO can be efficiently annotated, enriching these datasets. This can extend the lifespan of these older datasets, and can allow research to train models that require more fine-grain object annotations using datasets like COCO, which generally employ high-level semantic categories.

## E  COMPUTATIONAL REQUIREMENTS

All experiments in the paper can be replicated with 6,720 L40S GPU-hours, and models used in this work can be configured to fit on smaller GPUs with as few as 16GB of VRAM.

## F  EXPERIMENTAL SETUPS

In the following section, we present hyperparameters used in our method, and in the training process for object detection models from Section 4 in the main paper.

Training Hyperparameters for YOLO

- Model Size : 20M parameters (YOLOm variant)
- Epochs : 300

- Batch size : 16
- Image size : 640 pixels
- Optim : SGD
- LR : 0.01

BBox proposal

- Objectness threshold : 0.2
- NMS IoU threshold : 0.8

RAG Step 1

- K : 10 for PASCAL in-domain, 5 for ImageNet reasoning
- Secondary NMS IoU threshold : 0.5
- Minimum semantic IoU : 0.20 for PASCAL, 0.27 for ImageNet
- Minimum anchor image size : 0.30 for PASCAL, 0.50 for ImageNet

RAG Step 1

- Minimum number of anchor instances : 2
- Majority label fraction : 0.6 for PASCAL, varied for ImageNet

