# OpenReview forum: "Adaptively Labeling Vision Datasets Via Instance-Level Retrieval"
_ICLR.cc/2026/Conference — ICLR 2026 Conference Withdrawn Submission_

### Official Review · Reviewer_kvra · 2025-10-26

**Soundness:** 2
**Presentation:** 2
**Contribution:** 1
**Rating:** 2
**Confidence:** 3

**Summary:**

This paper proposes automatically constructing object detection datasets from large collections of unlabeled images. With only a handful of seed examples, the method discovers and labels relevant training data by comparing self-supervised object representations. First, candidate instances are mined across all unlabeled images using standard object detectors. Next, similarity between anchor instances and unlabeled candidates is measured with a Semantic Intersection over Union (Semantic IoU) computed from DINO2 features of the target object extracted by SAM2. Finally, high-quality candidates are filtered and returned as newly labeled instances for downstream training. Using the automatically labeled dataset, detection performance shows marginal improvement.

**Strengths:**

This paper attempts to develop an automated pipeline for generating detection datasets using high-performing models such as SAM2 and DINO2. The resulting dataset yields marginal improvements in detection model performance.

**Weaknesses:**

- The paper proposed a heuristic pipeline primarily leveraging DINO2 and SAM2 to automatically generate detection label.
- Performance gains of the detection models trained by the datasets generated by the proposed pipeline are marginal. The performance gain is possibly due to DINO2 and SAM2 knowledge transfer to the model, and it is unclear whether the proposed pipeline contributes the performance improvement..
- Automatic detection label generation is investigated priorly but the paper does not compare against existing methods such as
https://arxiv.org/html/2307.15710v2
https://github.com/autodistill/autodistill?tab=readme-ov-file#object-detection
https://github.com/IDEA-Research/Grounded-Segment-Anything
... (there are more)

**Questions:**

- How the encoded number of patches can be fixed N? The N size (can be varying) depends on the number of patch segmented by SAM2 which is dependent on the object size.

---

### Official Review · Reviewer_1u4k · 2025-10-31

**Soundness:** 2
**Presentation:** 3
**Contribution:** 1
**Rating:** 2
**Confidence:** 4

**Summary:**

The paper explores a pseudo-labeling setup for the object detection task. Images from the labeled part of the training set act as queries to find relevant and high-quality training samples of the same instance in the unlabeled set. In order to find candidate objects in the unlabeled set, an existing pre-trained detector is used. Subsequently, a pre-trained SAM model is used to create a mask for the relevant object for which DINOv2 features are extracted. Image-to-image similarity is then computed via the proposed semantic IoU metric between two sets of DINOv2 features. This metric involves a Hungarian matching to assign feature correspondences. The method is applied to the Pascal VOC dataset by considering part of the dataset as unlabeled.

**Strengths:**

**S1)** The need for massive, high-quality training datasets is evident from the current practices of foundation model pre-training. The curation of web-scraped data is a critical component in this process yet very often not publicly reported. This paper addresses an important research direction by proposing an automated method to identify and leverage relevant samples from large unlabeled data collection.

**Weaknesses:**

**W1)** Leveraging a smaller set of labeled images to pseudo-label a set of unlabeled images is one of the main directions used in semi-supervised learning. The paper lacks any discussion of how the method fits into this field and any comparisons to existing techniques. Several examples of the semi-supervised object detection line of work are:

[1] Unbiased Teacher for Semi-Supervised Object Detection, Liu et al., *ICLR 2021*

[2] Cross-Domain Adaptive Teacher for Object Detection, Li et al., *CVPR 2022*

[3] Detecting Twenty-thousand Classes using Image-level Supervision, *ECCV 2022*

[4] End-to-End Semi-Supervised Object Detection with Soft Teacher, Xu et al., *ICCV 2021*



**W2)** Similar to the previous point, the paper lacks any discussion about existing instance-level retrieval approaches. One of the main paper contributions, the Semantic IoU metric, is similar to prior work and should be compared against them. Works like Razavian et al. [5] use an analogous metric (also Chamfer similarity [6]) to estimate similarity between two sets of features. There are also learned methods, like AMES [7], which is trained on DINOv2 to estimate such similarity via a small transformer network.

[5] Visual Instance Retrieval with Deep Convolutional Networks, Razavian et al., *ITE Transactions on Media Technology and Applications 2016*

[6] Parametric correspondence and chamfer matching: Two new techniques for image matching, Barrow et al., *Image Understanding Workshop 1977*

[7] AMES: Asymmetric and Memory-Efficient Similarity Estimation for Instance-level Retrieval, Suma et al., *ECCV 2024*



**W3)** The whole pipeline includes several hyperparameters that are only vaguely mentioned in the text of the method. Their selected values are summarized in Chapter F of the Appendix, but it is not mentioned whether these values were selected based on the evaluation performance, or using some validation dataset. If the former is the case, then it is a weakness of the method since it cannot be applied to a different domain without tuning these parameters again.

**W4)** As the contribution of the paper is mostly a combination of existing pre-trained models, the applicability of the pipeline should be verified on more than one dataset. Furthermore, the approach is motivated by the under-leveraged "vast amounts of image data available on the internet", so the proposed retrieval approach should be applied on a large-scale unlabeled dataset. This would showcase the effectiveness of the retrieval under the existence of hard negative examples and verify its practical usability in terms of computational complexity.



Typos:

Line 187-188: Dinov2 lowercase inconsistent with the rest of the paper and repeated "N rows and D columns"

Line 51: two periods

mAP metric is typically reported in the range of 0 to 100.

**Questions:**

**Q1)** How are the hyperparameters, such as the NMS threshold, Objectness threshold, or Minimum semantic IoU selected?

---

### Official Review · Reviewer_D94N · 2025-11-01

**Soundness:** 2
**Presentation:** 3
**Contribution:** 2
**Rating:** 4
**Confidence:** 3

**Summary:**

The authors propose an adaptive labeling pipeline for object detection datasets. Specifically, the authors first use a pretrained region proposal network to extract candidate object instances from a large set of unlabeled images. The authors then represent each candidate with features from a self-supervised vision encoder. Finally, the authors retrieve the top-k most relevant candidates by nearest-neighbor search given a small set of seed images with bounding box annotations, which are then assigned labels based on their similarity to the seeds. Such a process yields an adaptively labeled dataset without additional manual annotations. Experimental results on in-domain and transfer labeling demonstrate the effectiveness of the proposed method.

**Strengths:**

-	The proposed method is well motivated. Adaptive data curation and labeling without human annotations is indeed important to meet evolving application demands.
-	The paper is generally well-written and easy to follow.
-	The experimental results seem promising.

**Weaknesses:**

-	I am concerned about the heavy computational costs induced by the proposed method, which could limit its real-world applications to some extent.
-	The authors use DINOv2 as a vision encoder to extract feature maps. It would be better to ablate other vision encoders as well (e.g., CLIP).
-	The authors only evaluate based on the YOLO detector. It would be better to provide results on several other popular detectors as well to demonstrate the generalizability of the proposed method.
-	What are the failure cases of the proposed method? Adding more analysis on failure cases will add more insights to the paper.

**Questions:**

See the questions mentioned above. Given the current status of the paper, I am leaning towards borderline reject and hope the authors could address my concerns during the rebuttal.

---

### Official Review · Reviewer_CSAL · 2025-11-03

**Soundness:** 2
**Presentation:** 3
**Contribution:** 2
**Rating:** 4
**Confidence:** 4

**Summary:**

This paper introduces a instance labeling method for object detection annotation generation. In detail, it adopts SAM2 model to extract instance mask and further utilize DINOv2 model to extract the mask related patch features. For unlabeled images, it first use Owlv2ForObjectDetection model to generate bounding boxes with objectness scores, after NMS, the remaining bounding boxes are send to DINOv2 model to get corresponding patch features. By comparing the patch features between the unlabeled boxes and ground truth annotations, a semantic IOU score are calculated to filtering highly similar instances. The similar instances are assigned the same label as the ground truth instance and then used for detector training. The generated pseudo label eventually achieves slightly better performance than original ground truth annotations.

**Strengths:**

1. The paper is well written and easy to follow.
2. The method produces good pseudo labels for object detection.

**Weaknesses:**

1. Lack of experiments. PASCAL VOC is a simple detection dataset, more experiments on more complex datasets should be conducted to strengthen the effectiveness of the proposed method. (e.g. COCO dataset)
2. Similar to the above concern, the PASCAL VOC dataset contain simple scenes with prominent and singular object features, which makes it more easy to generate pseudo labels. Also, ImageNet-1K is a image-classification dataset with objects in the image very obvious and can be easily detected. The method's generalization ability in complex images containing numerous objects could not be verified, reducing its effectiveness. More experiments on more complex images as unlabeled data are suggested.
3. SAM2 is a very strong segmentation model. Since it is used for mask generation, why not directly generate masks that corresponding to specific classes for pseudo label generation? For example, extract all the cats in the image, and directly generate bounding boxes according to the cat mask. Alternatively, SAM2 can be used to extract image features from the detection bounding box and semantic features of a specific category, and then the similarity can be calculated to generate pseudo-labels. The necessity of using semantic IoU to filter pseudo-labels cannot be proven.

**Questions:**

Refer to the weakness.

---

### Official Review · Reviewer_djvJ · 2025-11-12

**Soundness:** 2
**Presentation:** 2
**Contribution:** 2
**Rating:** 4
**Confidence:** 4

**Summary:**

The paper introduces Instance-Level Retrieval (ILR), a retrieval-augmented pipeline that starts from a handful of human-annotated “anchor” bounding boxes and automatically labels large pools of unlabeled images by ranking self-supervised object representations with a new “Semantic IoU” metric. On Pascal VOC, detectors trained with 1 k or 4 k adaptively retrieved boxes outperform same-size human-annotated subsets by ~0.08 mAP; when the pipeline is used to transfer VOC labels to ImageNet-1K images, a further +0.10 mAP gain is obtained. A dataset-distillation experiment shows that filtering the original VOC training set with Semantic IoU yields +0.04 mAP over random subsets. Extensive ablations and controlled scale-matched experiments are offered, together with a discussion of safeguards and ethical filters.

**Strengths:**

The paper is well written.

**Weaknesses:**

1.All experiments are confined to the 20 Pascal VOC classes; there is no evidence that the metric remains discriminative for fine-grained or long-tail categories where inter-class visual distance is smaller.
2.The method inherits any label noise or bias in the seed boxes; no analysis is given on how many anchors per class or what anchor diversity is minimally required, making real-world deployment risky when experts are unavailable.
3.Semantic IoU needs pairwise Hungarian matching between every candidate and every anchor; complexity is O(|anchors|×|candidates|×patches²), so scaling to web-scale unlabeled pools (billions of images) is impractical without additional approximate-search layers that are not explored.
4.The ImageNet transfer experiment still mixes retrieved samples with VOC originals in the best numbers; the purely adaptive rows (no VOC images) lag behind the mixed ones, indicating that the pipeline alone does not fully close the domain gap.

**Questions:**

1.How does performance change when anchors come from noisy web tags or only one bounding box per class instead of clean VOC annotations?
2.Could an approximate nearest-neighbor or inverted-index structure retain most of the mAP gain while reducing the O(N²) cost?
3.Have the authors tried fine-grained datasets (e.g., CUB-200) where subtle visual differences might break the patch-matching similarity?
4.What fraction of the retrieved ImageNet boxes are false positives when manually inspected, and does this vary by VOC class?
5.Does the pipeline work for non-rigid or heavily occluded objects where SAM2 masks and DINOv2 patches may be unreliable?

---

### Note · Authors · 2025-11-24

I have read and agree with the venue's withdrawal policy on behalf of myself and my co-authors.